# Promoting the health and wellbeing of children: A feasibility study of a digital tool among professionals

Magda Skogberg[1☉], Karolina Mackiewicz[2‡], Kristel Mänd[3‡], Lehte Tuuling[3‡], Indra Urdzina-Merca[4‡], Sanna Salanterä[1,5‡], Anni Pakarinen[1☉]*

1 Department of Nursing Science, University of Turku, Turku, Finland, 2 European Connected Health Alliance, Helsinki, Finland, 3 Rakvere City Government, Rakvere, Estonia, 4 Welfare Department, Jurmala City Council, Jurmal, Latvia, 5 Turku University Hospital, Turku, Finland

☉ These authors contributed equally to this work.
‡ KM, KM, LT, IUM and SS also contributed equally to this work.
* anni.pakarinen@utu.fi

**Data Availability Statement:** Our study's minimal underlying data set (SUS results) have been submitted as Supporting Information files (S1 and S2 Tables) after fully anonymized of the

## Abstract

The foundations of children's health and wellbeing are laid in early childhood. A gamified app (EmpowerKids tool) was designed to support professionals to have discussions with 6- to 12-year-olds from low-income families about their health and wellbeing. The aim of this feasibility study was to evaluate the usability and acceptability of the tool from the perspective of professionals in social, health and education settings. The study was conducted using a one-group post-test-only design. The usability data were collected using System Usability Scale and the acceptability data were collected using an open-ended questionnaire distributed to professionals (n = 24) in Estonia, Finland and Latvia. The data were collected during two phases. The tool was modified further on the basis of the results. The total usability scores were 82/100 (first testing) and 84/100 (second testing), indicating excellent usability. The answers related to acceptability were divided into four categories: suitability for the context; satisfaction and quality; attractiveness; modification needs. The professionals perceived that the tool helped them to build an overall picture of a child's health and wellbeing, and to gain information about the child's individual needs. The requirements for modification detected during the first testing were mostly related to difficulties with textual expressions and graphics. No major modification requirements were expressed during the second testing. The tool is considered feasible and may be used by professionals from different settings to support children's health and wellbeing. Further studies are needed to evaluate the effectiveness of the tool from the perspective of child outcomes.

## Introduction

Childhood is a sensitive formative period of a person's life, during which the socioeconomic status of their parents can have immediate and long-lasting adverse effects on the person's health and wellbeing [1]. Health inequalities and health problems in adulthood are more

respondents. The data including professional's open-ended answers, presented in the study are available from PI Anni Pakarinen, anni.pakarinen@utu.fi.

**Funding:** Interreg Central Baltic Award number: CB465 Recipient: EmpowerKids project consortium.

**Competing interests:** The authors have declared that no competing interests exist.

common among people who have grown up in a low-income or vulnerable family than for people who are from wealthier families [2, 3]. Low socioeconomic status has also been associated with disparities in health literacy [4, 5]. Health literacy refers to an individual's capacity to access, process, and appropriately interpret and act on essential health information and services to make health-related decisions [6, 7]. Subsequently, low health literacy is related to poorer health behaviors [8] and worse health outcomes [9]. Related term, mental health literacy refers to individual's knowledge and beliefs about mental health disorders that help in their recognition, prevention and management [10, 11]. Low levels of both health literacy types correlate with poorer mental [11, 12] and physical health, more use of healthcare than preventive care, and increased mortality [13].

Although, parents' decisions on health-related issues in the family will affect a child's health and wellbeing, children are also entitled to influence matters that concern their own life [14]. A child's wellbeing can be seen as a multi-dimensional construct, which consists of mental, physical and social dimensions [15]. Health behaviors, such as those relating to physical activity [16] and nutrition [17], in addition to family and personal resources [18, 19] and a balanced daily rhythm [20, 21], are factors that have an effect on a child's health and wellbeing. The foundations of a person's health, wellbeing and life success are laid in early childhood [22]; therefore, efforts to promote health and wellbeing should be targeted at this period of life [23].

Professionals working with young children in health and social care or educational settings are in a key position to promote health and wellbeing [24]. During interventions, it is fundamental to empower a child's active participation [24] and increase the child's knowledge, motivation, competence and self-efficacy [25]. Supporting an individual's ability to make health-promoting decisions in their own life may have a significant influence on their overall health and wellbeing in the long term [26]. Methods of promoting the health and wellbeing of children should be efficient and evidence-based [27]. The purpose should be to achieve optimal health and wellbeing, and to remove health inequalities; for example, by ensuring equal opportunities and resources for everyone [28]. These methods should also be child-centered, age-appropriate [29] and implemented in a way that supports the child's health literacy [8, 30].

Considering the child-centered methods, children of today are familiar with digital devices and applications already from the very early years of age [31]. To attract children with digital interventions, attention should be paid to the age-appropriate user interface and content [32, 33]. Schools offer an optimal setting for interventions among children and digital interventions seem to attract school-aged children [34]. According to a recent review, published in 2021, in school-based health promotive interventions, effective interventions included digital components, such as internet-based counselling, text messages, websites and social media. Results showed that these digital interventions often included also other components, such as face-to-face discussions with teachers and interventions addressed to parents [35]. Many of previous digital interventions address children's health behaviors, but there seems to be lack of digital interventions addressing child's mental, social health and wellbeing [34].

Health games can provide child-friendly and effective ways of promoting health [36, 37] and health education [38, 39]. Evidence suggests that health games and gamified apps have positive effects on children's health and wellbeing [40], their health behaviors [41, 42] and their health-related knowledge [43]. Despite being evidence-based, however, to be effective health games and gamified apps also need to be motivating, engaging and entertaining [44]. In addition, the content of the games should be age appropriate and ethically sustainable [45].

This study was conducted as part of a project entitled "EmpowerKids–Health education and social advice for low-income families with small children," which ran from the year 2016 to 2018 and was co-financed by the Central Baltic Programme 2014–2020. The EmpowerKids project focused on tackling social exclusion, inadequate health information and inadequate

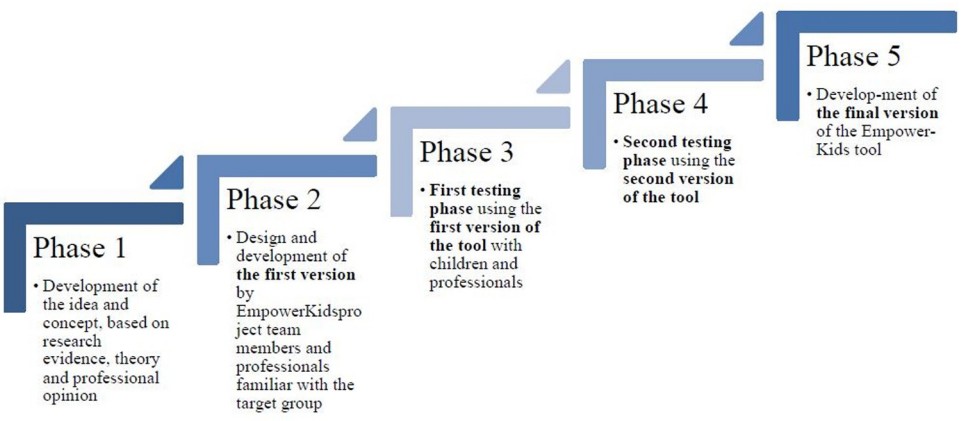

**Fig 1. Iterative user-centered design process.**

advice provided by social care services among children in low-income families in Estonia, Finland and Latvia. The project aimed to develop an intervention that would support professionals in promoting the health and wellbeing of children [46].

Given that games and apps are an integral part of children's everyday life, a gamified, Internet-enabled app (the EmpowerKids tool; also referred to as "the tool") was designed to support professionals in discussions with individual children about their health behaviors, physical activity and nutrition, family and personal resources, and daily habits. The EmpowerKids tool was developed at the University of Turku in collaboration with health professionals, social workers and preschool teachers from Estonia, Latvia and Finland [46, 47]. The development followed an iterative user-centered design process [48], which included five phases (Fig 1). Phases three and four cover the feasibility study reported in this article.

Feasibility studies are important in the development and testing of an intervention. A feasibility study explores the relevance, sustainability and modification requirements of an intervention. In general, such studies have different areas of focus (e.g., acceptability) depending on the intervention and the aim of the study. The chosen area of focus determines the optimal research design [49, 50].

The aim of this feasibility study was to evaluate the usability and acceptability of the EmpowerKids tool from the perspective of professionals in social, health and education settings.

## Materials and methods

### Study design

The study used a one-group post-test-only design [51]. The data were collected during the feasibility testing in February 2017 and again in September 2017 in order to examine professionals' opinions about usability and acceptability of the EmpowerKids tool (Fig 2).

**Intervention.** The target group for the intervention consisted of 6- to 12-year-olds from low-income families, and the professionals who were working with these children in social, health and educational settings. The intervention had two phases. First, a child entered information into the tool about their own health behaviors, their family and personal resources and their daily habits. Second, a health discussion took place between the child and a professional. The discussion with the child was based on the results provided by the tool for the child's physical activity, nutrition, family and personal resources and daily habits. The professionals were given a 1,5-hour training session about the intervention before they used it.

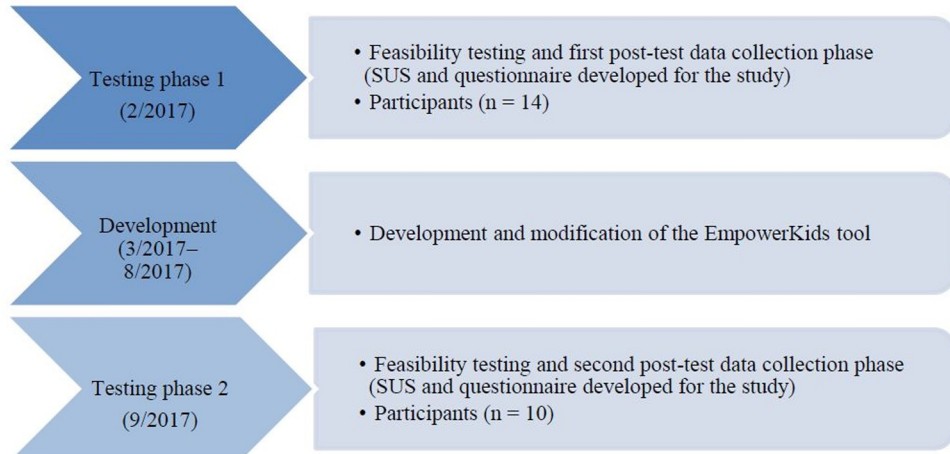

**Fig 2. Study design.**

The tool was designed with two user interfaces: one for the child (Fig 3) and one for the professional. It consisted of four sections: physical activity, nutrition, family and personal resources and daily rhythm. In the physical activity and nutrition sections (Fig 4), the child made choices based on their personal health behaviors by picking suitable apples (containing alternatives) for the baskets. In the daily rhythm section, the child made choices based on their personal habits (e.g., hygiene routines, hobbies, meeting friends), by picking suitable apples (containing alternatives) for the baskets. The baskets stood for the frequency of the habit ("every day," "a few times a week," "once a week," "sometimes"). In the resources section, the child scored different statements (n = 15) by choosing facial expressions that were based on a 4-point Likert scale (from "I agree" to "I disagree"). A 4-point Likert scale was chosen based on the views of the health professionals, social workers and preschool teachers. Literature among children has shown the reliability and validity of 4-point Likert scales [52]. The children were instructed to choose a face (from a smiley face to a sad face) that best described their level of agreement with each statement (e.g., "I feel safe," "I feel good at home," "I enjoy learning") by expressing strong or mild agreement or disagreement. The scale was developed based

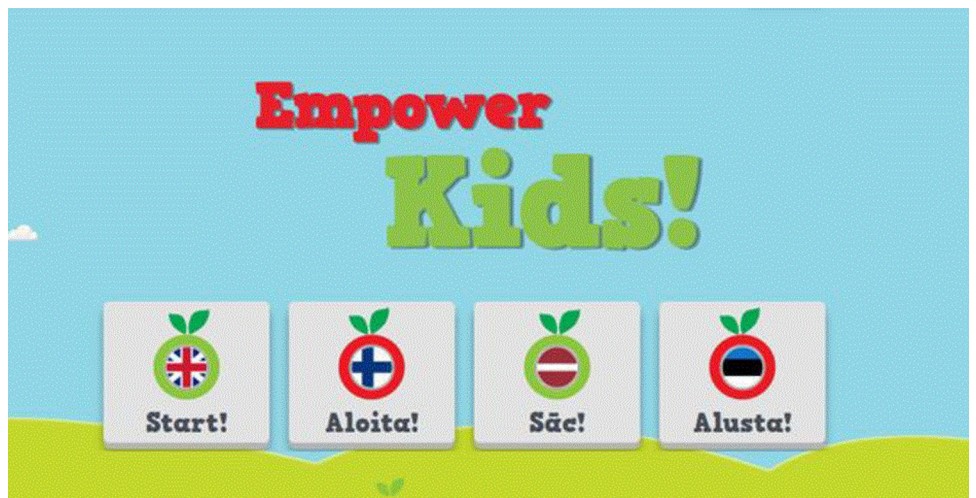

**Fig 3. Children's interface of the EmpowerKids tool.**

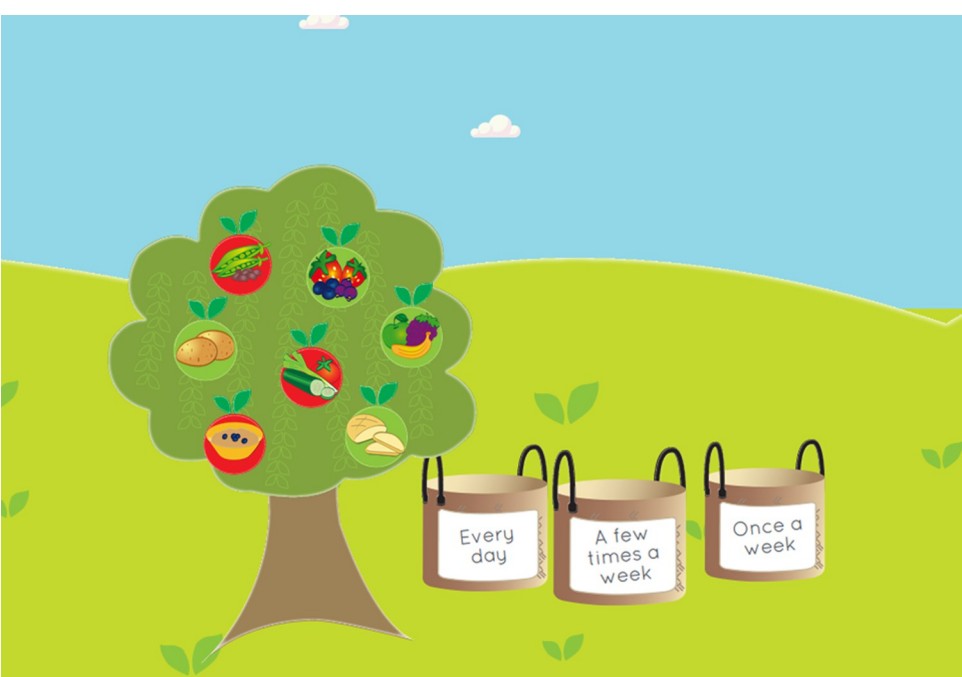

**Fig 4. Nutrition section of the EmpowerKids tool.**

on previous literature covering subjective and objective aspects of child wellbeing [e.g., 53–55] and involved experts in health research along with professionals working with children from the fields of health, social work and education. The set of statements included was proposed to represent the physical, emotional/psychological, social, educational and economic wellbeing of the child (Fig 5).

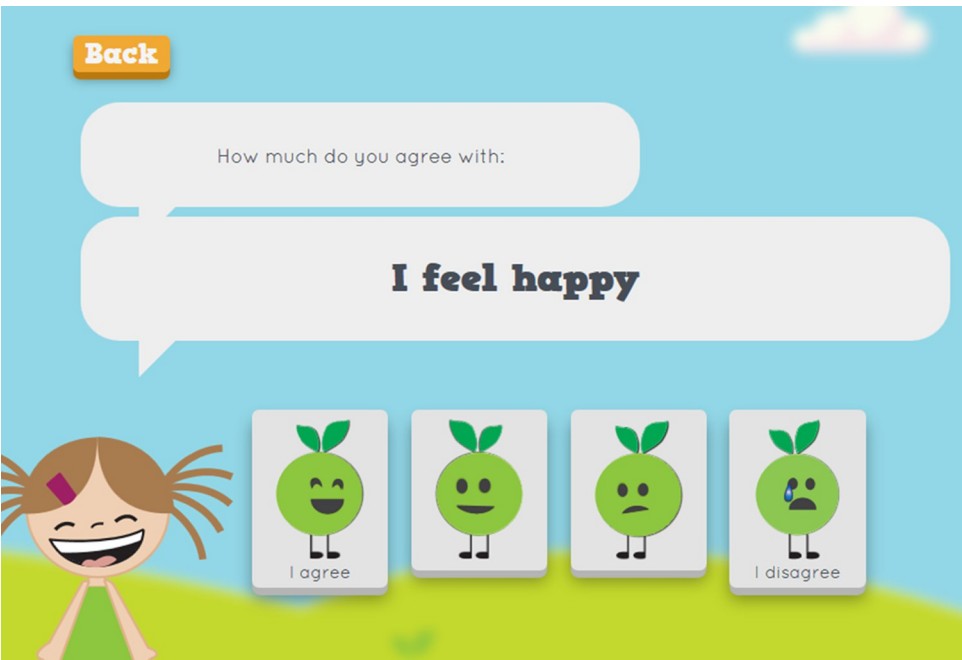

**Fig 5. Resources section of the EmpowerKids tool.**

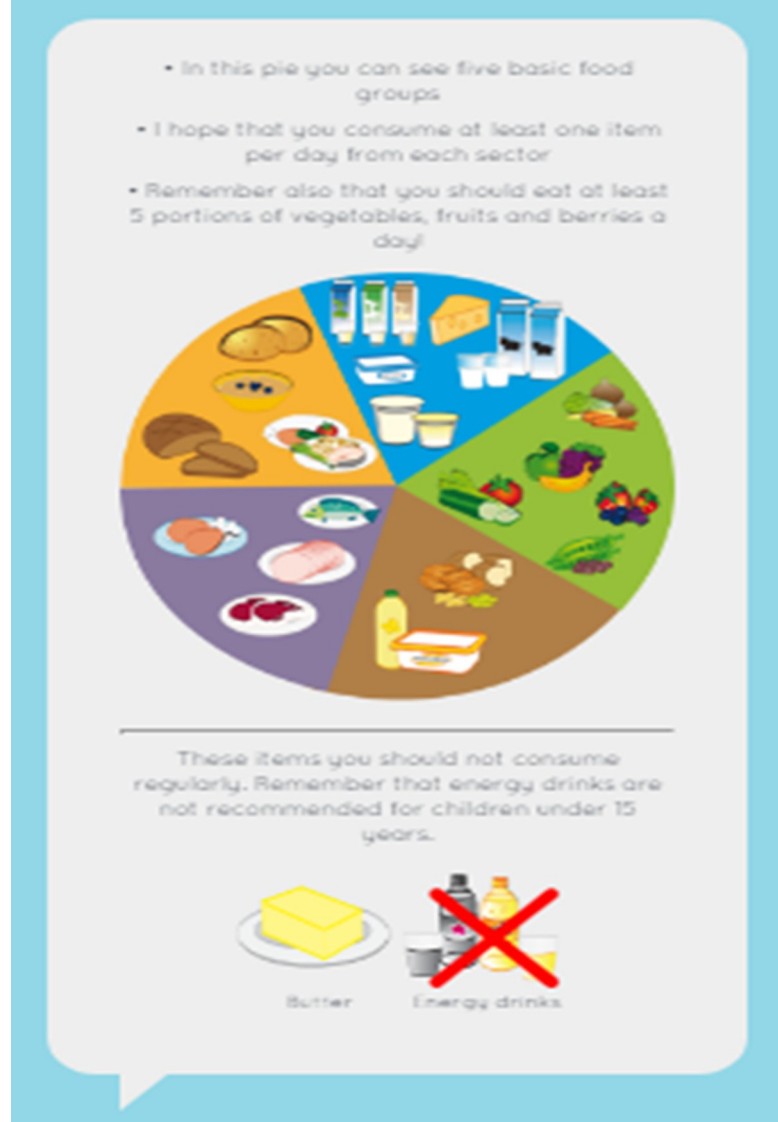

**Fig 6. Feedback for healthy nutrition.**

After completing each section, the children received visual feedback on their results. Feedback related to health behaviors was based on international [56] and national recommendations (i.e., current nutritional recommendations for children in each country).

The professionals also received the feedback on each child's results (Fig 6).

They could then use the visual feedback in their discussion with the child. The tool's database allowed for the collection of metadata about children's health and wellbeing, which can be used, for example, to guide the planning of activities and interventions in an organization.

## Study sample and recruitment

The study participants were recruited during the EmpowerKids project and the data were collected using a purposive sampling method. There was no predetermined sample size, since feasibility studies are not expected to have large sample sizes to show statistically significant

findings. Rather, the aim was to gain understanding about the intervention in real-life settings with various people that best represent the intended group of end-users [57, 58]. All the eligible professionals, attending to the intervention and using the tool with children, were recruited to the study. Professionals were given the information letter and informed consents were acquired. All together 14 professionals in the first testing phase and 10 in the second testing phase, voluntarily participated in the study. The study participants were professionals working with children with different roles in the organization.

During the first phase of testing (February 2017), the tool was tested in three countries (Estonia, Finland and Latvia) by children ranging from 6 to 12 years old (n = 95) and professionals working with these children in childcare (n = 11) and basic education settings (n = 3), with different roles in the organizations; manager (n = 1), head of the education (n = 1), project coordinator (n = 1), social worker (n = 3), elementary school teacher (n = 3), preschool teacher (n = 3), school health nurse (n = 1) and social pedagogue (n = 1). The study settings included school healthcare (n = 1), kindergartens (n = 4), children's centers or social centers (n = 5), elementary schools (n = 3) and a city welfare department (n = 1).

During the second phase of testing (September 2017), the tool was tested in three countries (Estonia, Finland and Latvia) by a different set of children ranging from 6 to 12 years old (n = 69) and different professionals working with these children in childcare (n = 9) and basic education settings (n = 1), with different roles in the organizations; manager (n = 2), head of the education (n = 2), project coordinator (n = 1), social worker (n = 3), elementary school teacher (n = 1), preschool teacher (n = 1). The study settings included kindergarten (n = 2), children center or social center (n = 5), elementary school (n = 3) and city welfare department (n = 1).

## Data collection and outcome measures

The data were collected using the System Usability Scale (SUS) [59] and an open-ended questionnaire after the professionals had used the EmpowerKids tool during their health discussions with children.

**System Usability Scale.** The SUS evaluates the subjective assessment of the usability of a specific system [60]. Evidence shows that the SUS [59] is a reliable and valid measure of usability [60]. SUS have been used among Finnish population before in many studies and the internal consistency of the SUS measured with the Cronbach's alpha has been from 0.831 to 0.89 [61, 62]. The SUS is a ten-item Likert scale with five response options that yield the respondents' level of agreement or disagreement with statements on the usability of the tool. The ten items in the scale alternate, with five positive statements (odd numbers) and five negative statements (even numbers). The score can range from 0 to 100, where a higher number represents a more usable tool [60]. The SUS used in this study was translated into Finnish, Estonian and Latvian.

**Acceptability of the tool.** A questionnaire was developed especially for this study (Table 1). The questionnaire evaluated the acceptability of the tool and modification needs.

**Table 1. Acceptability questionnaire developed for the study.**

| | |
|---|---|
| 1 | What did you like about the EmpowerKids app? |
| 2 | What did you dislike about the EmpowerKids app? |
| 3 | What do you want to be retained in the EmpowerKids app? |
| 4 | What do you want to be removed or changed in the EmpowerKids app? |
| 5 | Are there any other comments you want to make about the EmpowerKids app? |

Questions for the acceptability were based on previous literature on acceptability, where acceptability can be assessed based on the respondent's perceptions on the attractiveness, satisfaction, suitability and perceived appropriateness on the service, tool or device under evaluation [49]. The questionnaire consisted of five open-ended questions that assessed the respondents' opinions about the EmpowerKids tool. The questions covered general opinions on elements of the tool and ideas for further modification of the tool. First two questions targeted the respondent's perception on attractiveness and satisfaction, second two questions targeted the respondent's perception on suitability and perceived appropriateness, and last question targeted the respondent's perception on modification needs of the tool. The respondents were also able to add any other comments they had about using the tool.

**Descriptive statistics.** A standard analysis of the average SUS scores and the synthesis of the results was performed using a template in Microsoft Excel 2016 and standard deviations for each item mean value were based on Standard deviation calculations. The overall SUS score was calculated according to the special calculating system: for each positive answer (statements 1, 3, 5, 7, 9), one point from the user's response (1 to 5) was subtracted and for each negative answer (statements 2, 4, 6, 8, 10), the user's response (1 to 5) was subtracted from five. The sum of the scores for all ten items was then multiplied by 2,5, which gave an overall SUS score, ranging from 0 to 100 [55]. Complementary adjective ratings for the SUS scores were also defined to initiate a distinct description of the overall usability of the EmpowerKids tool [63].

**Qualitative data analysis.** The responses to the open-ended questions were analyzed after the first phase of testing by performing an inductive content analysis to produce a synthesis of the main findings [64]. The aim was to attain a condensed and comprehensive description of the acceptability of the EmpowerKids tool. During the analysis, we moved from specific to general data, so that specific items were observed and then combined into a general category. The outcome of the inductive content analysis were four generic categories; suitability for the context, satisfaction and quality, attractiveness, and modification needs. The generic categories describe the main categories for which professionals expressed an opinion on the usability and acceptability of the tool. The responses to the same open-ended questions from the second phase of testing were analyzed by performing a deductive content analysis, using the category structure that was created from the previous inductive content analysis. This made it possible to compare the categories between the first and second testing points.

## Ethical considerations

The authors assert that all procedures in the study were performed in accordance with the ethical principles and responsible conduct of research [65, 66]. This study was conducted among adult participants (professionals in the given settings) during a EmpowerKids project. The children and their parents, who were taking part in the project, were informed about the project and consents were acquired from them before start of the project in their schools or other settings. All the eligible professionals were given the information letter and informed consents were acquired. Professionals provided their written informed consents before participation and voluntarily participated in the study. This consent procedure was approved by the Ethics committee of University of Turku and is in line with the ethical review practice in Finland.

## Results

### Usability of the EmpowerKids tool

**First phase of testing (version 1).** The overall average usability of the EmpowerKids tool in all countries was good, with an average SUS score of 80.14 (SD±15.45) across all countries.

**Table 2. SUS scores after first testing phase.**

| Item | FIN (n = 3) | | | EE (n = 7) | | | | | | | LV (n = 4) | | | | Mean ± SD (FIN) | Mean ± SD (EE) | Mean ± SD (LV) | Mean ± SD (all) |
|---|---|---|---|---|---|---|---|---|---|---|---|---|---|---|---|---|---|---|
| 1. I think that I would like to use this app frequently. | 3 | 3 | 3 | 3 | 3 | 3 | 3 | 4 | 5 | 5 | 4 | 4 | 4 | 4 | 3 ± 0.00 | 3.7 ± 0.95 | 4 ± 0.00 | 3.64 ± 0.75 |
| 2. I found this app unnecessarily complex. | 4 | 2 | 4 | 1 | 2 | 2 | 4 | 1 | 1 | 1 | 1 | 1 | 1 | 1 | 3.33 ± 1.15 | 1.71 ± 1.11 | 1 ± 0.00 | 1.85 ± 1.23 |
| 3. I thought this app was easy to use | 2 | 4 | 4 | 4 | 4 | 4 | 3 | 5 | 4 | 5 | 5 | 5 | 4 | 4 | 3.33 ± 1.15 | 4.14 ± 0.69 | 4.5 ± 0.58 | 4.07 ± 0.83 |
| 4. I think that I would need assistance to be able to use this app. | 1 | 2 | 2 | 1 | 1 | 1 | 1 | 1 | 1 | 1 | 1 | 1 | 1 | 1 | 1.66 ± 0.58 | 1 ± 0.00 | 1 ± 0.00 | 1.14 ± 0.36 |
| 5. I found the various functions in this app were well integrated. | 4 | 4 | 3 | 3 | 3 | 3 | 4 | 3 | 5 | 5 | 4 | 4 | 5 | 5 | 3.66 ± 0.58 | 3.7 ± 0.95 | 4.5 ± 0.58 | 3.92 ± 0.83 |
| 6. I thought there was too much inconsistency in this app. | 3 | 2 | 2 | 1 | 2 | 2 | 3 | 2 | 1 | 1 | 1 | 1 | 1 | 1 | 2.33 ± 0.58 | 1.71 ± 0.76 | 1 ± 0.00 | 1.64 ± 0.74 |
| 7. I would imagine that most people would learn to use this app very quickly | 4 | 3 | 4 | 4 | 4 | 5 | 5 | 5 | 5 | 5 | 5 | 5 | 5 | 5 | 3.66 ± 0.58 | 4.71 ± 0.49 | 5 ± 0.00 | 4.57 ± 0.65 |
| 8. I found this app very cumbersome/awkward to use. | 2 | 2 | 2 | 1 | 2 | 3 | 4 | 2 | 1 | 1 | 1 | 1 | 1 | 1 | 2 ± 0.00 | 2 ± 1.15 | 1 ± 0.00 | 1.71 ± 0.91 |
| 9. I felt very confident using this app. | 4 | 4 | 4 | 5 | 5 | 4 | 4 | 5 | 5 | 5 | 5 | 5 | 5 | 5 | 4 ± 0.00 | 4.71 ± 0.49 | 5 ± 0.00 | 4.64 ± 0.50 |
| 10. I needed to learn a lot of things before I could get going with this app. | 1 | 3 | 4 | 1 | 2 | 2 | 2 | 1 | 1 | 1 | 1 | 1 | 1 | 1 | 2.66 ± 1.53 | 1.43 ± 0.53 | 1 ± 0.00 | 1.57 ± 0.94 |

SD = Standard deviation.

The professionals' average system usability ratings were higher in Latvia (95, SD±0.00) than in Estonia (82.86, SD±13.65) and Finland (64.17 SD±3.82) (Tables 2 and 3). For the professionals in Estonia (n = 7), the scores ranged from 62,5 (OK) to 100 (best imaginable). For the professionals in Finland (n = 3), the scores ranged from 60 (OK) to 67,5 (OK). All the professionals in Latvia (n = 4) rated the tool at 95 (excellent).

**Second phase of testing (version 2).** The overall average usability of the EmpowerKids tool in all countries was good, with an average SUS score of 86.67 (SD±12.33) across all countries (Tables 4 and 5). The professionals' average system usability ratings were higher in Finland (95, SD±0.00) and Latvia (92.50, SD±6.45) than in Estonia (74.00, SD±8.40). The professional in Finland (n = 1) rated the usability of the tool at 95 (excellent). For the

**Table 3. Total SUS scores and adjective ratings after first testing phase.**

| Country | Total score | SUS score | Adjective rating |
|---|---|---|---|
| **Estonia (n = 7)** | 34 | 85 | Excellent |
| | 30 | 75 | Good |
| | 29 | 72,5 | Good |
| | 25 | 62,5 | Ok |
| | 35 | 87,5 | Excellent |
| | 39 | 97,5 | Excellent |
| | 40 | 100 | Best imaginable |
| **Finland (n = 3)** | 26 | 65 | Ok |
| | 27 | 67,5 | Ok |
| | 24 | 60 | Ok |
| **Latvia (n = 4)** | 38 | 95 | Excellent |
| | 38 | 95 | Excellent |
| | 38 | 95 | Excellent |
| | 38 | 95 | Excellent |

**Table 4. SUS scores after second testing phase.**

| Item | FIN (n = 1) | EE (n = 5) | | | | | LV (n = 4) | | | | Mean ± SD (FIN) | Mean ± SD (EE) | Mean ± SD (LV) | Mean ± SD (all) |
|---|---|---|---|---|---|---|---|---|---|---|---|---|---|---|
| 1. I think that I would like to use this app frequently. | 3 | 4 | 3 | 4 | 3 | 4 | 4 | 5 | 5 | 5 | 3 ± 0 | 3.6 ± 0.55 | 4.75 ± 0.50 | 4 ± 0.82 |
| 2. I found this app unnecessarily complex. | 1 | 1 | 4 | 2 | 2 | 1 | 1 | 1 | 2 | 2 | 1 ± 0 | 2 ± 1.22 | 1.5 ± 0.58 | 1.7 ± 0.95 |
| 3. I thought this app was easy to use | 5 | 5 | 3 | 5 | 4 | 4 | 5 | 5 | 4 | 5 | 5 ± 0 | 4.2 ± 0.83 | 4.75 ± 0.50 | 4.5 ± 0.71 |
| 4. I think that I would need assistance to be able to use this app. | 1 | 1 | 1 | 1 | 2 | 1 | 1 | 1 | 1 | 2 | 1 ± 0 | 1.2 ± 0.45 | 1.25 ± 0.50 | 1.2 ± 0.42 |
| 5. I found the various functions in this app were well integrated. | 5 | 4 | 5 | 4 | 3 | 3 | 5 | 5 | 4 | 4 | 5 ± 0 | 3.8 ± 0.84 | 4.5 ± 0.58 | 4.2 ± 0.79 |
| 6. I thought there was too much inconsistency in this app. | 1 | 3 | 4 | 2 | 3 | 3 | 2 | 1 | 1 | 1 | 1 ± 0.00 | 3 ± 0.71 | 1.25 ± 0.50 | 2.1 ± 1.10 |
| 7. I would imagine that most people would learn to use this app very quickly | 5 | 4 | 5 | 4 | 4 | 5 | 5 | 5 | 5 | 4 | 5 ± 0.00 | 4.4 ± 0.55 | 4.75 ± 0.50 | 4.6 ± 0.52 |
| 8. I found this app very cumbersome/awkward to use. | 1 | 4 | 4 | 2 | 3 | 2 | 1 | 1 | 1 | 1 | 1 ± 0.00 | 3 ± 1.00 | 1 ± 0.00 | 2 ± 1.25 |
| 9. I felt very confident using this app. | 5 | 5 | 5 | 5 | 3 | 4 | 5 | 5 | 4 | 4 | 5 ± 0.00 | 4.4 ± 0.89 | 4.5 ± 0.58 | 4.5 ± 0.71 |
| 10. I needed to learn a lot of things before I could get going with this app. | 1 | 1 | 1 | 3 | 2 | 1 | 1 | 1 | 1 | 2 | 1 ± 0.00 | 1.6 ± 0.89 | 1.25 ± 0.50 | 1.4 ± 0.70 |

SD = Standard deviation.

professionals in Latvia (n = 4), the scores ranged from 85 (excellent) to 100 (best imaginable). For the professionals in Estonia (n = 5), the scores ranged from 62,5 (OK) to 80 (good).

## Acceptability of the EmpowerKids tool

**First phase of testing.** The overall acceptability of the tool was based on its suitability for the context (children aged 6–12 years from low-income families, and professionals), satisfaction and quality, and attractiveness. The requirements for modification and other comments regarding the use of the tool were related to its overall coherence. The abbreviations (EE, FI, LV) used in the following sections stand for Estonia (EE), Finland (FI) and Latvia (LV).

*Suitability for the context.* The professionals considered that the tool was easy to use and simple to explain to children (EE, FI). The game design, graphics and music were also thought to be appropriate for children (EE, FI). In the professionals' opinion, the child's mood and desires (EE) and the current season (e.g., winter) might influence the child's gameplay and response (EE, FI). Of the tool's four games (physical activity, nutrition, daily rhythm and resources), the resources game was thought to be the easiest for a child to play independently.

**Table 5. Total SUS scores and adjective ratings after second testing phase.**

| Country | Total score | SUS score | Adjective rating |
|---|---|---|---|
| **Estonia (n = 5)** | 32 | 80 | Good |
| | 27 | 67,5 | Ok |
| | 32 | 80 | Good |
| | 25 | 62,5 | Ok |
| | 32 | 80 | Good |
| **Finland (n = 1)** | 38 | 95 | Excellent |
| **Latvia (n = 4)** | 38 | 95 | Excellent |
| | 40 | 100 | Best imaginable |
| | 36 | 90 | Excellent |
| | 34 | 85 | Excellent |

This was because children responded to the questions by choosing from four easy-to-understand "smiley faces." The physical activity and nutrition games contained some graphs, which were thought to be irrelevant or confusing for players (EE, FI, LV). Furthermore, the professionals commented that the feedback after every section was too long and not tailored enough to a child's individual needs (EE).

*Satisfaction and quality*. The professionals in all three countries agreed that the tool was excellent for use in discussions with children. The concept of the tool was also appreciated, in that it offered professionals a new way to promote children's health and wellbeing. The professionals valued its comprehensive overview of a child's health and wellbeing was valued (EE, FI, LV). Technical difficulties, such as slowness and incorrect language options, were reported by professionals from one of the countries (EE). A few professionals also described the tool as time-consuming, due to all four sections having to be played at once (EE, FI).

*Attractiveness*. The game graphics and music (birdsong) used in the tool was described as enjoyable and pleasant (EE, FI). The professionals expressed that children were eager to play the game and that they wanted to play it again later (EE). Some of the graphics were hard to understand (EE, FI); nevertheless, the professionals expressed that the tool was interesting for children to play and talk about afterwards (EE, LV).

*Modification needs*. The professionals in all three countries were happy with the tool, but they made some suggestions for modification. They required more answer options in the daily rhythm game to cover choices such as the weekend and particular seasons (EE, FI, LV). Some of the professionals would have preferred simpler or fewer answer options in the physical activity and nutrition games (EE, FI). Professionals in every country expressed that the graphics should be clear enough for a child to understand (EE, FI, LV). In addition, they mentioned that capital letters would be a better option than lower-case letters for younger children (EE). The option to pause a single game and continue later would also have been appreciated (EE). Apart from these suggestions, one professional put forward the idea of including a "talking owl" to provide in-game help; this would enable the children to play some sections more independently, without needing constant assistance from an adult (FI).

**Second phase of testing.** The EmpowerKids tool was modified in line with the relevant requirements that were observed in the first testing phase. The main modifications included adding more activity options for the daily rhythm game, adding the option to change the text to capital letters, and making it possible to play only one section at one time. In the second testing phase, the overall opinions regarding the acceptability of the tool were mostly concerned with the practicalities of using the tool in childcare settings.

*Suitability for the context*. All four sections of the EmpowerKids tool were found to be necessary for the professionals to be able to build an overall picture of a child's health and wellbeing (EE, FI, LV). In addition, all the sections were important sources of information for detecting a child's individual needs (EE, FI, LV). However, for the tool to be useful for a child, the child has to understand that they are meant to evaluate their current health status and health behavior. According to the professionals, this was a challenge for children of preschool age, some of whom answered according to what they wanted to be true or what they craved for. In addition, the large number of answer options to choose from in the tool as a whole, and the number of sports options in the physical activity section in particular, were thought to be exhausting for the children. This was because several alternatives had to be explained to children who had not experienced them or did not know what they were (EE, LV) or who found them confusing because of language barriers (FI). The professionals thought this was time-consuming (EE, LV). The professionals valued having discussions with a child during the gameplay, because this gave them an opportunity to talk with the child about health and wellbeing while they were playing and choosing alternatives EE). On the other hand, the

professionals seemed to find discussion during the gameplay time-consuming and tiring for the younger children, whose attention sometimes wandered (EE, LV). The feature that made it possible to play the sections separately, and not necessarily in the same order, was an appreciated modification (LV).

*Satisfaction and quality*. All the professionals expressed that all four sections of the tool were needed (EE, FI, LV). The sections that received the highest satisfaction scores were daily rhythm (EE), nutrition (LV) and resources (EE. LV). The professionals appreciated the immediate feedback provided by the tool, which they could use in their discussions with a child about their health and wellbeing (EE). The tool was thought to be easy to use (FI) and a suitable method for use in individual conversations with children (EE). The tool was described as a useful method of working with children and as a resource that could tie in with various theme days (LV). There were some technical issues (such as slowness) in some parts of the tool (LV).

*Attractiveness*. The EmpowerKids tool was described as an interesting (LV), fun, enjoyable and cheerful (EE) application for children to use. The game design was described as easy and understandable, and the game graphics were described as attractive (FI). Using the app on a tablet was thought to be an easy way to start a discussion with the children in a different sort of way (LV).

*Modification needs*. The professionals suggested that the physical activity section could be modified by reducing or combining the sports options and by making this section shorter (EE, LV). They also suggested reducing the number of answer options in all sections, to improve understanding of the tool among children of preschool age (EE). Technical improvements regarding the speed and providing an alternative option for the appearance of the clock in the daily rhythm section were also mentioned (EE). In addition, it was suggested that the feedback section after every section could be more tailored to the individual child (FI).

## Discussion and conclusions

In this feasibility study, our aim was to explore the acceptability and usability of the EmpowerKids tool for professional users in various childcare and basic education settings. The findings of this study support the usability and acceptability of the tool in Estonia, Finland and Latvia. The results indicate that the usability of the tool is good and that the user experience is acceptable among professionals. The professionals perceived the tool as an excellent discussion resource to use with children. They valued the comprehensive overview provided of a child's health and wellbeing for use in their discussions.

Children's wellbeing encompasses quality of life in a broad sense [67]. It can be seen as a multi-dimensional construct that consists of mental, physical and social dimensions [15]. Approaching children's wellbeing from a subjective perspective gives children an opportunity to express their own wellbeing and avoids making decisions in advance about which dimensions of life are important to children and the weights of the different dimensions [68]. The EmpowerKids tool aims to assess children's health and wellbeing in different dimensions; when used as part of an intervention, it offers a feasible tool for professionals to use in order to have health discussions with children in a child-centered way. The tool not only focuses on the child's challenges but also tries to identify the child's strengths so professionals can draw on these to tackle the challenges detected as posing a risk to the child's health and wellbeing [15, 69].

Games have been shown to have potential in health education for children, particularly in promoting physical activity and healthier dietary habits [37]; meanwhile, children perceive games as more useful and beneficial than other digital educational activities [70]. Schools offer

one viable setting for health-promoting and game-based interventions. When implemented in a primary school context, a game-based intervention was shown to be effective in improving knowledge about mental health among 9- to 11-year-olds and encouraging them to put problems into perspective [71]. Another game-based intervention promoted positive attitudes and self-efficacy in relation to healthy eating among 10- to 11-year-olds [72]. However, these previous game-based interventions concentrated on a narrow dimension (mental vs physical) of a child's wellbeing. Thus, the EmpowerKids tool is more multi-dimensional than previous game-based interventions, which calls for rigorous evaluation in school settings.

There is a need to monitor the wellbeing of children using systematic tools [69, 73]. The EmpowerKids tool offers one way of assessing the subjective wellbeing of children. Digital wellbeing assessment tools enable fluent and real-time data collection, alongside analyses and longitudinal monitoring of users' health data; they also offer an easy access database for professionals, organizations and decision makers to use in order to guide the planning of targeted and needs-based interventions and services for children [74]. There are few existing validated web-based tools for assessing children's wellbeing, but being more "structured query" type measures [69], they differ from the EmpowerKids tool, which offers a more subjective overview of a child's wellbeing for use in health discussions. Thus, more research is needed before the data from the tool can be used as valid measurements and as a database that can inform, for example, decision makers.

The EmpowerKids tool is promising in terms of detecting a child's individual health and wellbeing needs. It is a useful resource for professionals in discussions about children's health behaviors, their overall wellbeing and their support needs. The tool is most suitable for children of school age; children of preschool age and children with cognitive impairments are likely to need more assistance to understand the concept and content of the tool. In addition, the low socioeconomic status and insufficient previous knowledge of the children who participated in this study might have influenced their understanding of some of the content, as indicated by the professionals. Thus, the tool should be modified further to fit better with every socioeconomic context and support the health literacy of children and families from lower socioeconomic groups.

One of the strengths of this feasibility study was the data collection methods, which included quantitative and qualitative assessments of the feasibility of the EmpowerKids tool. The usability and acceptability were measured twice in diverse contexts in three different countries. Furthermore, the selected measure for assessing usability has been widely used and is considered valid [59, 75]. This study focused on the perceptions of the professionals. Even though the professionals also reflected on the perceptions of children in their answers, it would be useful to evaluate the feasibility of the tool from the perspective of children. It would also be meaningful to evaluate the effectiveness of the intervention from the perspective of child outcomes, especially before planning its implementation [50].

There were also some limitations to this study. The sample size and sampling method were restricted to professionals who were working on the EmpowerKids project. However, in comparison with colleagues outside the project, the chosen study participants might have had a stronger interest in developing methods for professionals working on health and wellbeing with children from low-income families. Nevertheless, the generalizability of the results is limited.

The results of this feasibility study suggest that the EmpowerKids tool is feasible in several childcare and education settings in Estonia, Finland and Latvia. Children's age and level of cognitive understanding, professional users' previous knowledge of each child, and suitable education and training in using the tool are important factors to consider when using this method with children. When used as part of an intervention, the tool offers a promising

method of identifying a child's individual circumstances and needs in order to promote their health and wellbeing.

## Supporting information

**S1 Table.**
(PDF)

**S2 Table.**
(PDF)

## Acknowledgments

We would like to thank our colleagues at the EmpowerKids project for their support in planning, organizing and implementing the study in the three countries. We also express our profound gratitude to the children, and to the study participants, for their time and generous feedback.

## Author Contributions

**Conceptualization:** Karolina Mackiewicz, Kristel Mänd, Lehte Tuuling, Indra Urdzina-Merca, Sanna Salanterä, Anni Pakarinen.

**Data curation:** Magda Skogberg, Anni Pakarinen.

**Formal analysis:** Magda Skogberg, Anni Pakarinen.

**Investigation:** Anni Pakarinen.

**Methodology:** Anni Pakarinen.

**Project administration:** Anni Pakarinen.

**Supervision:** Anni Pakarinen.

**Validation:** Anni Pakarinen.

**Writing – original draft:** Magda Skogberg, Karolina Mackiewicz, Kristel Mänd, Lehte Tuuling, Indra Urdzina-Merca, Sanna Salanterä, Anni Pakarinen.

**Writing – review & editing:** Magda Skogberg, Karolina Mackiewicz, Kristel Mänd, Lehte Tuuling, Indra Urdzina-Merca, Sanna Salanterä, Anni Pakarinen.

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
