## [Decision Letter · Decision Letter 0]

2 Dec 2021

PONE-D-21-30852Promoting the health and wellbeing of children: a feasibility study of a digital tool among professionalsPLOS ONE

Dear Dr. Pakarinen,

Thank you for submitting your manuscript to PLOS ONE. After careful consideration, we feel that it has merit but does not fully meet PLOS ONE’s publication criteria as it currently stands. Therefore, we invite you to submit a revised version of the manuscript that addresses the points raised during the review process.

We look forward to receiving your revised manuscript.

Kind regards,

Prabhat Mittal, Ph.D.

Academic Editor

PLOS ONE

Journal Requirements:

" ext-link-type="uri" xlink:type="simple">https://journals.plos.org/plosone/s/file?id=ba62/PLOSOne_formatting_sample_title_authors_affiliations.pdf"

a) Did participants provide their written or verbal informed consent to participate in this study?

3. Please include your tables as part of your main manuscript and remove the individual files. Please note that supplementary tables (should remain/ be uploaded) as separate "supporting information" files.

4. In the Methods section of the manuscript please amend your current ethics statement to confirm that your named ethics committee specifically approved or waived this study.

Additional Editor Comments:

Refer to the following research  

Chakraborty, P., Mittal, P., Gupta, M. S., Yadav, S., Arora, A. (2021). Opinion of students on online education during the COVID-19 pandemic. Human Behavior and Emerging Technologies, 3(3), 357–365. https://doi.org/10.1002/hbe2.240

Yadav, S., Chakraborty, P., Mittal, P., Arora, U. (2018). Children aged 6–24 months like to watch YouTube videos but could not learn anything from them. Acta Paediatrica, International Journal of Paediatrics, 107(8), 1461–1466. https://doi.org/10.1111/apa.14291

Yadav, S., Chakraborty, P., Mittal, P. (2021). User Interface of a Drawing App for Children: Design and Effectiveness. In Advances in Intelligent Systems and Computing (Vol. 1165, pp. 53–61). https://doi.org/10.1007/978-981-15-5113-0_4

Yadav, S., Chakraborty, P., Mittal, P. (2021). Designing Drawing Apps for Children: Artistic and Technological Factors. International Journal of Human-Computer Interaction, 1–15. https://doi.org/10.1080/10447318.2021.1926113

Reviewers' comments:

Reviewer's Responses to Questions

**Comments to the Author**

1. Is the manuscript technically sound, and do the data support the conclusions?

Reviewer #1: Yes

Reviewer #2: Yes

2. Has the statistical analysis been performed appropriately and rigorously? 

Reviewer #1: N/A

Reviewer #2: No

3. Have the authors made all data underlying the findings in their manuscript fully available?

Reviewer #1: Yes

Reviewer #2: Yes

4. Is the manuscript presented in an intelligible fashion and written in standard English?

Reviewer #1: Yes

Reviewer #2: Yes

5. Review Comments to the Author

Reviewer #1: The study aimed at evaluating the usability and acceptability of the tool (EmpowerKids tool) from the perspective of professionals in social, health and educational settings. Based on these, it is of paramount benefit to the entire populace globally. The Authors should verify the use of British and US English. Citations should be duly done.

Reviewer #2: Dear Author,

The manuscript has been well draft but at the same time there are certain observations that i would like to share:

1. The survey instrument has been developed for this research based on literature. As it is a developed scale it needs to be checked for reliability and validity. Expert opinion has been captured as presented in the manuscript but the it is required to summarize the outcome in the manuscript. The scales capturing the constructs needs to be tested and the Cronbach alpha for each construct needs to be quoted in the design section of the manuscript.

2. The constructs to test the feasibility of the instrument have arrived form literature as mentioned in the manuscript. A conceptual framework in the manuscript can further bring clarity to the paper.

3. Appreciate the work done across geographies. The rationale for selection of Latvia etc needs to be justified.

4. Statistical analysis: Mere presentation of mean will not be sufficient. An in depth analysis is required.

5. The statistical outcomes need to be strengthened. The outcomes presented are mere basics.

6. Using smilys to capture data is a good idea. But Likert is a 5 point scale. Only 4 options are presented. Justify with adequate citation.

6. PLOS authors have the option to publish the peer review history of their article (what does this mean?). If published, this will include your full peer review and any attached files.

Reviewer #1: **Yes: **YAHAYA ABDULLAHI

Reviewer #2: **Yes: **Smitha Nayak

---

## [Author Response · Author response to Decision Letter 0]

16 Feb 2022

Comments with responses:

https://journals.plos.org/plosone/s/file?id=ba62/PLOSOne_formatting_sample_title_authors_affiliations.pdf"

Manuscript has now been checked to meet the PLOS ONE's style requirements

a) Did participants provide their written or verbal informed consent to participate in this study?

Ethics statement has now been amended in the Methods section.

3. Please include your tables as part of your main manuscript and remove the individual files. Please note that supplementary tables (should remain/ be uploaded) as separate "supporting information" files.

Tables has now been included as part of the manuscript and individual files have been removed. 

4. In the Methods section of the manuscript please amend your current ethics statement to confirm that your named ethics committee specifically approved or waived this study.

Ethics statement has now been amended in the Methods section.

5. In your Data Availability statement, you have not specified where the minimal data set underlying the results described in your manuscript can be found. PLOS defines a study's minimal data set as the underlying data used to reach the conclusions drawn in the manuscript and any additional data required to replicate the reported study findings in their entirety. All PLOS journals require that the minimal data set be made fully available. Upon re-submitting your revised manuscript, please upload your study’s minimal underlying data set as either Supporting Information files or to a stable, public repository and include the relevant URLs, DOIs, or accession numbers within your revised cover letter. Any potentially identifying patient information must be fully anonymized.

We have now uploaded our study’s minimal underlying data set after fully anonymzed the respondents and included the data in the Supporting Information files (S1 and S2).

Yadav, S., Chakraborty, P., Mittal, P., Arora, U. (2018). Children aged 6–24 months like to watch YouTube videos but could not learn anything from them. Acta Paediatrica, International Journal of Paediatrics, 107(8), 1461–1466. https://doi.org/10.1111/apa.14291

Yadav, S., Chakraborty, P., Mittal, P. (2021). User Interface of a Drawing App for Children: Design and Effectiveness. In Advances in Intelligent Systems and Computing (Vol. 1165, pp. 53–61). https://doi.org/10.1007/978-981-15-5113-0_4

Yadav, S., Chakraborty, P., Mittal, P. (2021). Designing Drawing Apps for Children: Artistic and Technological Factors. International Journal of Human-Computer Interaction, 1–15. https://doi.org/10.1080/10447318.2021.1926113

Thank you for introducing related and recent literature that is applicable to be referred in this manuscript. We have referred to above-mentioned research in our manuscript.

5. Review Comments to the Author

Reviewer #1: The study aimed at evaluating the usability and acceptability of the tool (EmpowerKids tool) from the perspective of professionals in social, health and educational settings. Based on these, it is of paramount benefit to the entire populace globally. The Authors should verify the use of British and US English. Citations should be duly done.

Thank you for the comment. The language of this manuscript has been verified with official translation agency using British English. Citations have now been double checked for consistency of the Journal instructions 

Reviewer #2: Dear Author,

The manuscript has been well draft but at the same time there are certain observations that i would like to share:

1. The survey instrument has been developed for this research based on literature. As it is a developed scale it needs to be checked for reliability and validity. Expert opinion has been captured as presented in the manuscript but the it is required to summarize the outcome in the manuscript. The scales capturing the constructs needs to be tested and the Cronbach alpha for each construct needs to be quoted in the design section of the manuscript.

Thank you for the comment. The scale has been used in studies among Finnish population before and we have now extended the description of the validity and reliability of the scale by adding description on the Cronbach alpha of the scale into the methods section. The internal consistency of the SUS measured with the Cronbach's alpha in Finnish population has been high.

2. The constructs to test the feasibility of the instrument have arrived form literature as mentioned in the manuscript. A conceptual framework in the manuscript can further bring clarity to the paper.

Thank you for the comment. We have not described enough detailed how the constructs of the feasibility instrument were developed. We have now elaborated more the conceptual framework of the feasibility in the methods section. 

3. Appreciate the work done across geographies. The rationale for selection of Latvia etc needs to be justified.

Thank you for the comment. This study was conducted during the EmpowerKids project, which was a cross-border collaboration project between stakeholders in Finland, Estonia and Latvia. Project addressed social exclusion and inadequate health information and social advice among the children. Thus, the reason for choosing above-mentioned countries was that the intervention was implemented in the participating countries and enabled the collection of data from the professionals participating to the intervention. 

4. Statistical analysis: Mere presentation of mean will not be sufficient. An in depth analysis is required.

Thank you for the comment. Analyses have been made according to the instructions by the SUS developer, but we have now included also the standard deviations to the analyses and described more detailed in the results section and in Tables 2 and 4. Adjective ratings are based on mainly used 

5. The statistical outcomes need to be strengthened. The outcomes presented are mere basics.

Thank you for the comment. We have now strengthened the statistical outcomes as described above.

6. Using smilys to capture data is a good idea. But Likert is a 5 point scale. Only 4 options are presented. Justify with adequate citation.

Thank you for the comment. We discussed on the type Likert-scale with the expert team during the development of the EmpowerKids app. While our children users were from age six to age 13, we ended up to a decision to use 4 point Likert scale instead of 5 point Likert scale. This decision was based on the expert opinions and experiences with children, but also to the literature, for example Alan Kabasakal (2020). According to their study: “Children can use 3- and 4-point Likert-type scales, reliability coefficient increased with an increasing number of response options for the scale and validity of 3- and 4-point versions of the scale were appropriate and 2-point version was weak.” The use of 4 option smily scale has been now justified in the intervention description.

---

## [Editor Report · Decision Letter 1]

1 Mar 2022

Promoting the health and wellbeing of children: a feasibility study of a digital tool among professionals

PONE-D-21-30852R1

Dear Dr. Pakarinen,

We’re pleased to inform you that your manuscript has been judged scientifically suitable for publication and will be formally accepted for publication once it meets all outstanding technical requirements.

Kind regards,

Prabhat Mittal, Ph.D.

Academic Editor

PLOS ONE
---

## [Editor Report · Acceptance letter]

8 Mar 2022

PONE-D-21-30852R1 

Promoting the health and wellbeing of children: a feasibility study of a digital tool among professionals 

Dear Dr. Pakarinen:

I'm pleased to inform you that your manuscript has been deemed suitable for publication in PLOS ONE. Congratulations! Your manuscript is now with our production department. 

Kind regards, 

on behalf of

Dr. Prabhat Mittal 

Academic Editor

PLOS ONE